# Computational Eurotransplant kidney allocation simulations demonstrate the feasibility and benefit of T-cell epitope matching

**Matthias Niemann** [1] *, **Nils Lachmann** [2], **Kirsten Geneugelijk**[3], **Eric Spierings**[3]

**1** PIRCHE AG, Berlin, Germany, **2** Center for Tumor Medicine, H&I Laboratory, Charité University Medicine Berlin, Berlin, Germany, **3** Center of Translational Immunology, UMC Utrecht, Utrecht, The Netherlands

* matthias.niemann@pirche.com

**Data Availability Statement:** The results are based upon are publicly accessible datasets. The

## Abstract

The EuroTransplant Kidney Allocation System (ETKAS) aims at allocating organs to patients on the waiting list fairly whilst optimizing HLA match grades. ETKAS currently considers the number of HLA-A, -B, -DR mismatches. Evidently, epitope matching is biologically and clinically more relevant. We here executed ETKAS-based computer simulations to evaluate the impact of epitope matching on allocation and compared the strategies. A virtual population of 400,000 individuals was generated using the National Marrow Donor Program (NMDP) haplotype frequency dataset of 2011. Using this population, a waiting list of 10,400 patients was constructed and maintained during simulation, matching the 2015 Eurotransplant Annual Report characteristics. Unacceptable antigens were assigned randomly relative to their frequency using HLAMatchmaker. Over 22,600 kidneys were allocated in 10 years in triplicate using Markov Chain Monte Carlo simulations on 32-CPU-core cloud-computing instances. T-cell epitopes were calculated using the www.pirche.com portal. Waiting list effects were evaluated against ETKAS for five epitope matching scenarios. Baseline simulations of ETKAS slightly overestimated reported average HLA match grades. The best balanced scenario maintained prioritisation of HLA A-B-DR fully matched donors while replacing the HLA match grade by PIRCHE-II score and exchanging the HLA mismatch probability (MMP) by epitope MMP. This setup showed no considerable impact on kidney exchange rates and waiting time. PIRCHE-II scores improved, whereas the average HLA match grade diminishes slightly, yet leading to an improved estimated graft survival. We conclude that epitope-based matching in deceased donor kidney allocation is feasible while maintaining equal balances on the waiting list.

## Author summary

Kidney transplantation is the best treatment option for patients suffering permanent loss of kidney function. High degrees of histocompatibility between patients and organ donors improve long-term function of transplanted kidneys. In order to ensure fair access to

respective repositories are are referred to in the applied research protocol: dx.doi.org/10.17504/protocols.io.bqrtmv6n.

**Funding:** The author(s) received no specific funding for this work.

**Competing interests:** I have read the journal's policy and the authors of this manuscript have the following competing interests: MN is an employee of PIRCHE AG that runs the PIRCHE web-portal. The UMC Utrecht has filed a patent application on the prediction of an alloimmune response against mismatched HLA. ES is listed as inventor on this patent. The authors of this manuscript NL and KG have no competing interests to disclose as described by PLOS Computational Biology.

**Abbreviations:** AM, Acceptable Mismatch; AUC, area under the curve; DSA, donor specific HLA antibody; ESP, Eurotransplant Senior Program; ETKAS, Eurotransplant kidney allocation system; ETKASPIR, PIRCHE-modified ETKAS; HLA, human leukocyte antigen; KDPI, kidney donor profile index; KPSAM, UNOS Kidney-Pancreas Simulation Allocation Model; MCMC, Markov chain Monte Carlo; MMP, mismatch probability; NMDP, National Marrow Donor Program; NT patient, not-transplantable patient; OPTN, Organ Procurement and Transplantation Network; PIRCHE, predicted indirectly recognizable HLA epitopes; PIRCHE-II RP, PIRCHE-II Risk Profile; PRA, panel reactive antibodies; SRTR, Scientific Registry of Transplant Recipients; SWML, sum of all distribution-weighted mean log(PIRCHE-II) values; T patient, transplantable patient; UNOS, United Network for Organ Sharing; UKAM, UNOS Kidney Allocation Model; WML, distribution-weighted mean log (PIRCHE-II).

transplantation whilst maximising utility of each donor kidney, organ allocation organizations established recipient waiting lists and well-balanced algorithms to allocate donors to patients. Changing the allocation algorithms requires careful consideration of side-effects to avoid disadvantages of certain groups of patients. In this study, we evaluated the feasibility of modifying the existing Eurotransplant Kidney Allocation System (ETKAS) to incorporate indirect T-cell epitope matching, a novel technique for assessing functional histocompatibility. Using Markov chain Monte Carlo simulations, we compared the modified allocation to the current algorithm and found an overall improvement of indirect T cell epitope compatibility. Simultaneously, we observed no negative impact on allocation fairness or waiting times. Our simulation framework may serve as a basis to evaluate further adjustments to ETKAS in the future. From our results, we conclude that epitope matching can be safely incorporated into ETKAS.

## Introduction

Organ transplantation represents the golden treatment standard for patients with permanent organ failure. A major complication of organ transplantation is the development of immune responses directed towards human leukocyte antigen (HLA) mismatches between donor and recipient followed by allograft rejection [1–3]. HLA matching and HLA antibodies are key factors in most allocation algorithms to avoid these responses [4,5]. Since previous research has shown that better HLA matching indeed resulted in a reduced need for immunosuppressive treatment as well as less allograft rejection [5], organ exchange organizations, including Eurotransplant, have implemented the HLA matching factor in their allocation strategy [6,7]. These strategies all aim for transplanting kidneys with low numbers of HLA mismatches between donors and recipients [5]. Although this approach is an effective method to reduce the risk for kidney allograft rejection, it has some limitations. First, for some patients, it may be more difficult to find donors with a low number of HLA mismatches due to the ethnic background of the patient. Second, not all HLA mismatches contribute equally to alloreactivity, as their immunogenicity may differ [8,9]. As such, kidney allocation via the Eurotransplant Kidney Allocation System (ETKAS) aims to establish an acceptable HLA match distribution and optimal overall transplant success rate, while shortening the waiting times, adjusting for infrequent HLA antigens and homozygosity thereof, and maintaining a balanced kidney exchange rate among countries [10].

ETKAS is the largest kidney allocation program of Eurotransplant with around 70% of the donors being allocated by it [11]. Currently, ETKAS considers the number of serological HLA-A, -B, -DR mismatches as a factor in their scoring system. However, in recent years, the concepts of B-cell and T-cell epitope matching in solid organ transplantation have evolved (reviewed in [12]). Epitope matching is based on structural and functional properties of the HLA proteins rather than belonging to the same serological group, which is the classical method of HLA matching. Various software tools were developed to model the epitope matching concepts for antibody epitopes [13–15] and T-cell epitopes (reviewed in [16]). The enhanced understanding on the immunobiology of alloreactive responses in solid organ transplantation leading to graft rejection, formed the basis of estimating the immunogenicity of HLA mismatches [17]. As the exact amino acid sequence of HLA antigens is available, algorithms are now in place to indicate which donor-specific HLA fragments (epitopes) presented by recipient HLA may in the cellular alloreactive response after solid organ transplantation (reviewed in [18]). Therefore, one of the approaches to estimate the clinical impact of

individual HLA mismatches on alloreactivity may be to quantify the total T-cell epitope load between donor and recipient [19,20]. Thus, the total T-cell epitope load may provide more information on the potential clinical consequences of HLA mismatches. These observations have led to a concept called epitope-based HLA matching (short: "epitope matching"): a matching strategy in which epitopes from HLA are matched.

The T-cell epitopes can be predicted using the PIRCHE algorithm. The PIRCHE algorithm considers peptides of donor HLA origin as being presented in the context of recipient HLA, followed by recognition by recipient T cells (reviewed in [16]). This model can predict indirect CD4+ T-cell recognition of allo-HLA derived peptides [16,17], a concept which is highly involved in graft failure after solid organ transplantation (review by Siu *et al.* [21]). Recent studies have shown that predicted donor-HLA derived epitopes presented on HLA class-II molecules, designated as PIRCHE-II, are related to HLA antibody formation [19,22]. Moreover, other retrospective organ transplantation studies using the PIRCHE algorithm have shown that T-cell epitope matching indeed leads to an improved transplant outcome [20,23]. Thus, these studies suggest that epitope matching may be biologically and clinically more relevant than simply counting the number of antigen matches and mismatches [21].

Since increased compatibility by applying epitope matching likely leads to a better outcome, incorporation of these techniques into the allocation systems may lead to a generally improved transplant outcome [24–26]. However, given donor organs are a scarce resource, it is questionable to what cost this matching improvement is achieved. ETKAS is the result of a well-balanced scoring system that was fine-tuned over the years [6,10,27,28]. Due to this fine-tuning, the allocation system is complex and the effects of changes are difficult to predict. Thus, when modifying an allocation system, it is crucial to prospectively evaluate performance and potential side effects systematically and thoroughly to rule out unwanted side effects. The value of computer simulations has shown its value in this context [27].

In the present study, we simulated Eurotransplant's largest kidney allocation program to evaluate the impact of including T-cell epitope matching. To this end, HLA-related factors in ETKAS were converted to PIRCHE-dependent factors. These PIRCHE-dependent factors were optimized in this study to achieve a high epitope matching result, which is expected to improve kidney graft survival of transplanted patients. The feasibility of our approach was concluded by observing a small impact on non-HLA outcome parameters, such as waiting time, waiting list size, and country balance.

## Materials and methods

### Creating a virtual population

Before initiation of the simulation procedures, a virtual waiting list was composed. For that purpose, we generated a virtual population of 400,000 individuals using the NMDP haplotype frequency dataset of European Caucasians (EURCAU) of 2011 [29]. Assuming random mating, haplotypes were combined according to their reported frequency. This virtual population served as a virtual donor pool and was used to populate the virtual waiting list (Fig 1).

### Populating the virtual waiting list

Initial virtual waiting lists were constructed from this virtual population by bootstrapping, matching the size and characteristics of the Eurotransplant Annual Report 2015 [11]. First, 10,400 virtual individuals and their HLA typings were randomly selected and removed from the virtual population. Subsequently, for the non-HLA typing characteristics, the parameters country, urgency, recipient age, blood group, percentage of PRA, and waiting time were extracted from the Eurotransplant Annual Report 2015 [11] and assigned to the virtual waiting

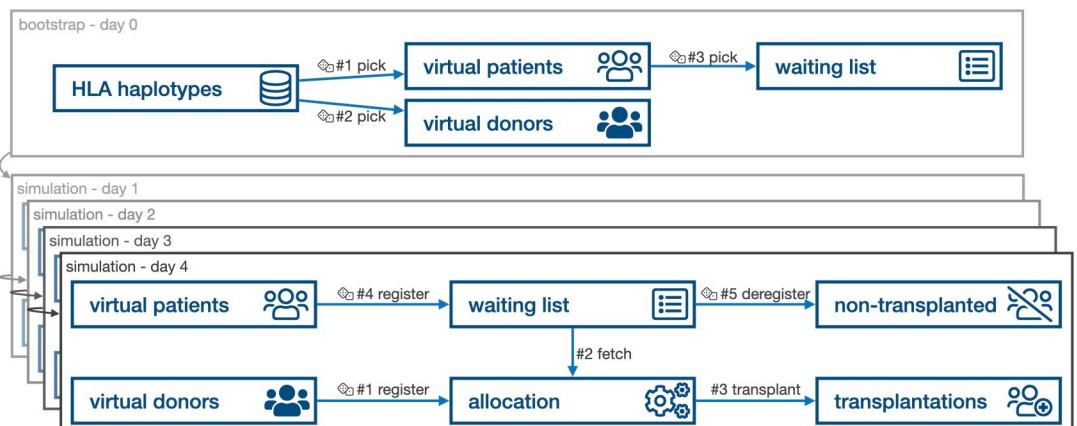

**Fig 1. Simulation schematic.** During the bootstrapping process, virtual patients and donors were created based on haplotype frequencies. The initial waiting list was randomly populated matching the distributions provided by Eurotransplant. During the simulation, waiting patients were randomly removed from the waiting list (e.g. unfit for transplantation, died waiting) and new patients were randomly registered to the waiting list. Virtual donors were randomly registered and allocated to the current waiting list resulting in simulated transplantations.

list population accordingly. Country, urgency and age were considered as independent parameters and thus assigned randomly, using the reported overall frequencies. Distributions for blood group, PRA, and waiting time were considered conditional to the virtual individuals' country and thus assigned according to country-dependent frequencies.

The publicly available data of the Eurotransplant Annual Report 2015 do not provide a distribution function of the waiting time of patients at time of registration to the waiting list. Therefore, the given intervals (0–1 years, 2–4 years, 5 years and longer) and their frequency were considered as continuous uniform distributions. The last interval was assumed to end at 9 years.

For virtual recipients with a PRA >0%, unacceptable antigens were assigned randomly relative to their population frequency. For each of those virtual recipients, unacceptable antigens were defined by taking antibody epitopes as defined by HLAMatchmaker [13] into account, designated as eplets. Initially, the frequencies of eplets present in the population's genotypes were calculated. For each virtual recipient, a number of random eplets not present in the set of self-eplets were selected that approximately summed up to the percentage of the assigned PRA. These eplets were translated into antigens that were considered as unacceptable for the recipient. Given each eplet's individual frequency, target PRA levels may be reached with low or high eplet counts.

## Basic allocation principles

After constructing a virtual donor pool and virtual initial waiting lists, allocations were simulated using a MCMC method. This simulation was applied iterating over time, considering each day in the allocation system as a single step (Fig 1). The core of the implemented simulation was a Gibbs sampler [30] that considered the virtual population's characteristics to create a sequence of transplantation events and waiting list snapshots. The resulting sequence allows estimating the distribution of match grades and allocation-relevant characteristics in virtual transplantations. Opposedly, the dynamics of the waiting list characteristics were observed.

For each step, we randomly selected and removed a virtual donor and its HLA typing from the virtual population. Subsequently, country of origin, donor type, blood group, and number of kidneys were assigned randomly, matching the distributions provided by the ET Annual

report 2015 [11]. In this procedure, the country of origin of the donor was assigned independently, whereas the number of kidneys per donor, donor type (i.e. cardiac versus brain death), and blood group were considered being donor-country dependent. These characteristics were added to the selected donor for simulation purposes and analyses.

In total, an average of 6.2 kidneys per day were allocated within the simulated ETKAS. Patients deregistered from the waiting list without being transplanted within ETKAS (e.g. death on waiting list, recovered, transplanted by another allocation program) were selected randomly conditional on the patient country with around 10.5 events per day. Registration of patients to the waiting list followed the same procedure as the waiting list bootstrapping with around 16.7 events per day (Eurotransplant Annual Report 2015 [11]). The virtual waiting list only contained actively waiting patients. The ET urgency status "not transplantable" was not considered during simulations.

A detailed protocol of the initialization, the bootstrapping and the simulation steps is available online at https://dx.doi.org/10.17504/protocols.io.bqrtmv6n

### PIRCHE-II and eplet calculations

In all simulations, PIRCHE-II were calculated as described before [19], using the PIRCHE web service (version 3.1, IMGT 3.36.0). The HLA-A, -B, -C, -DRB1 and -DQB1 loci were considered as peptide sources and virtual individuals' DRB1 as peptide presenters. Eplet matching, which was used to assign unacceptable mismatches to the virtual recipients (see above), considered eplet definitions by HLAMatchmaker version 2.0 (http://www.epitopes.net, downloaded August 2015) and was carried out as interlocus set difference of virtual donor and virtual individual.

### Computational infrastructure

The initialization and simulations were implemented in the Python programming language (Python Software Foundation, version 3.5.1). Simulation runs were executed three times for ETKAS. All PIRCHE-II based simulations were run once and the optimal PIRCHE-II based scenario (ETKASPIR-E) was repeated two more times. Each simulation was freshly initialized with a newly generated waiting list to rule out biased bootstrapping taking effect on the overall results. As the PIRCHE-II based allocation simulations were computationally intense, parallelization of PIRCHE-II calculations was implemented to reduce overall runtime. All simulations were executed on Amazon Web Services r4.8xlarge Elastic Cloud Compute instances (Amazon Web Services Inc., Seattle, US) with 32 compute cores and 244 GB of main memory. Each individual PIRCHE-II based allocation simulation took approximately 8 days of continuous computing with over 33 million PIRCHE-II score calculations.

### ETKAS allocation simulation

We first performed an allocation in triplicate using the unmodified ETKAS procedure as described in the Eurotransplant ETKAS Manual (Fig 2) [28], assuming an HLA genotype-independent outflow of donors to the AM program. These ETKAS simulations were used as a baseline simulation to validate the allocation simulations and to evaluate the effects of the PIRCHE-II modified allocation scenarios. In the ETKAS simulation, the waiting list was first filtered by donor eligibility (donor type allowed in the recipient's country) and blood group identity. Second, the waiting list was prioritized by full HLA match (A and B at serological broad level, DR at split level), homozygosity status and descending by point score. The simulated point score system strictly followed the ETKAS point score, thus including waiting time, urgency, country balance, distance between donor center and transplant center, HLA A-B-DR

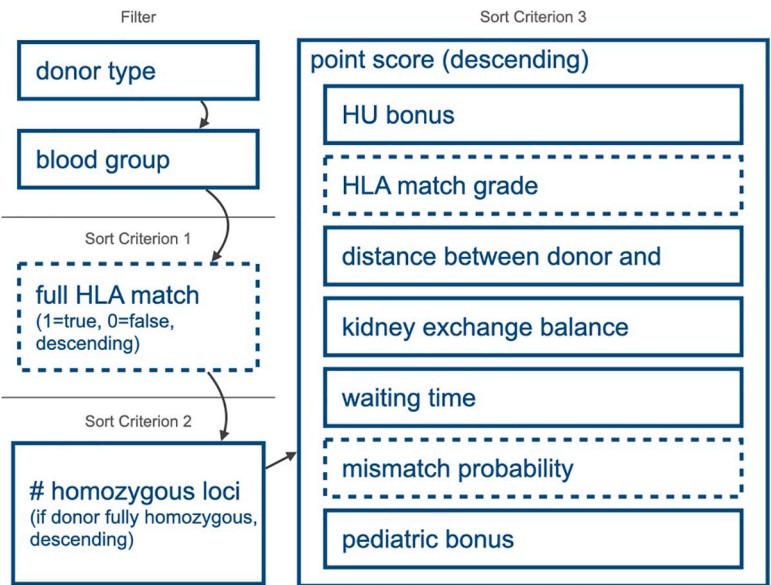

**Fig 2. ETKAS algorithm.** The basic ETKAS allocation consists of filtering steps and a cascade of sorting criteria. The dashed boxes refer to components involving HLA compatibility. These components were part of the altered allocation simulations. Other factors were unaltered.

match grade, and MMP. Pediatric patients received an additional one-time bonus. While allocating virtual donors, unacceptable mismatches defined for the recipient were avoided.

The likelihood and exact reason of an organ offer being rejected by the transplant center is undocumented. Certainly, there is a strong regional effect, as center policy and local waiting list demographics differ. To diminish unfounded assumptions about this regional variation, a constant probability (p = 0.1) of denying an offer was implemented. This inclusion means there is a 10% chance a kidney is refused for an individual patient across all countries, match grades and age groups.

All of these criteria mentioned above were implemented in the applied MCMC simulations. The spatial resolution of the allocation was limited to a national basis, as center specific data were not available. As a consequence, distance between donor center and patient center always preferred national pairs and did not further distinguish regional or local equivalent pairs. All allocation rules and parameters remained unchanged during simulations.

## Substituting HLA matching by PIRCHE-II based matching

In the PIRCHE-II modified allocation simulations, we evaluated the effects of replacing the HLA-based ETKAS scoring components by PIRCHE-II based equivalents, while leaving all the implemented non-HLA ETKAS parameters and scoring untouched. In these modifications, the total score and prioritization related to each individual scoring component remained unchanged and only the distribution of these points was modified (Table 1). These scores involved the full HLA match prioritization, HLA match grade, and mismatch probability. For all simulations and for the reference PROCARE dataset [20], the frequency of virtual transplantations depending on PIRCHE-II score was shown in density plots and the percentages for each of the four previously reported risk strata [19] were calculated.

**Full HLA match prioritization.** ETKAS prioritizes full HLA-matched (broad antigen level) donor-recipient combinations. In the PIRCHE-II based allocation simulations this prioritization was changed to those combinations resulting in a PIRCHE-II score lower than 9.

**Table 1. Description of the allocation simulation models.**

| Model | HLA identical | Points by match grade | Mismatch probability | Number of simulations |
|---|---|---|---|---|
| ETKAS | ET, Full HLA match priority | ET, based on HLA match grade | ET | 3 |
| ETKASPIR-A | PIRCHE < 9 | ET, based on HLA match grade | ET | 1 |
| ETKASPIR-B | ET, Full HLA match priority | PIRCHE, based on PIRCHE strata[19] | ET | 1 |
| ETKASPIR-C | ET, Full HLA match priority | PIRCHE, relative to negative exponential PIRCHE score | ET | 1 |
| ETKASPIR-D | ET, Full HLA match priority | PIRCHE inverse linear related to PIRCHE score, capped at 90 | ET | 1 |
| ETKASPIR-E | ET, Full HLA match priority | PIRCHE inverse linear related to PIRCHE score, capped at 90 | MMP-fitted PIRCHE RP polynomial | 3 |
| ETKASPIR-F | PIRCHE < 9 | PIRCHE inverse linear related to PIRCHE score, capped at 90 | MMP-fitted PIRCHE RP polynomial | 1 |

PIRCHE-II scores below 9 are considered very low [19] and should thus represent an optimal PIRCHE-II matched situation in kidney transplantation. This prioritization was executed in an identical fashion as in the current ETKAS prioritization for full HLA-A, -B, and -DRB1 matched combinations [28]. As such, donor-recipient combinations with a score below 9 were prioritized in the resulting ETKASPIR-A and -F models, regardless of the total score of the recipient on the waiting list.

**HLA match grade replacements.** We subsequently modified the ETKAS simulations by replacing HLA matching-based scoring parameters with PIRCHE-II based parameters. ETKAS assigns a maximum of 400 points depending on the HLA match (Fig 2). In the PIRCHE-II based allocation simulations, these 400 points were distributed depending on the PIRCHE-II score of a specific donor-recipient combination in three different ways (Fig 2 and Table 1). The first model used a categorization of each potential donor-recipient combination according to the previously defined ranges [19]. In this ETKASPIR-B model, matching scores were assigned as follows: PIRCHE-II <9: 400 points; PIRCHE-II 9–35: 266 points; PIRCHE-II 35–90: 133 points; PIRCHE-II > 90: 0 points. The second variant, implemented in the ETKASPIR-C model, applied a weighted negative exponential way, based on a continuous PIRCHE-II scoring formula, i.e. $score = 400 \times (0.958 \times e^{-0.032 \times PIRCHE})$. The third variant, implemented in the ETKASPIR-D, ETKASPIR-E and ETKASPIR-F models, included a linear distribution of the 400 points in the range of 0 to 90, i.e. $score = 400 \times \left(\frac{-1}{90} \times (PIRCHE + 1)\right)$, and zero points for all combinations with a PIRCHE-II score above 90 (Fig 3A).

**HLA mismatch probability.** ETKAS implemented the HLA MMP to create a more balanced opportunity in the allocation process for patients that are usually difficult to match over those easily being well-matched. Analogous to this system, the PIRCHE-II Risk Profile (PIRCHE-II RP) considers the likelihood of a patient being well-matched from the perspective of the PIRCHE-II score. To calculate the PIRCHE-II RP, we considered the most frequent 2011 NMDP EURCAU haplotypes [29] to build a set of 1199 unique virtual genotypes with their estimated population frequencies. Each of these genotypes was PIRCHE-II-matched with each of the individuals of the virtual population, leading to a total of over 479x10^6 PIRCHE-II calculations. Subsequently, the PIRCHE-II scores of the results per virtual individual were aggregated as a weighted median considering the virtual genotype frequency forming the PIRCHE-II RP median. Thus, the score indicates that half of the donor population is expected to have a lower PIRCHE-II score and half of them to have a higher PIRCHE-II score.

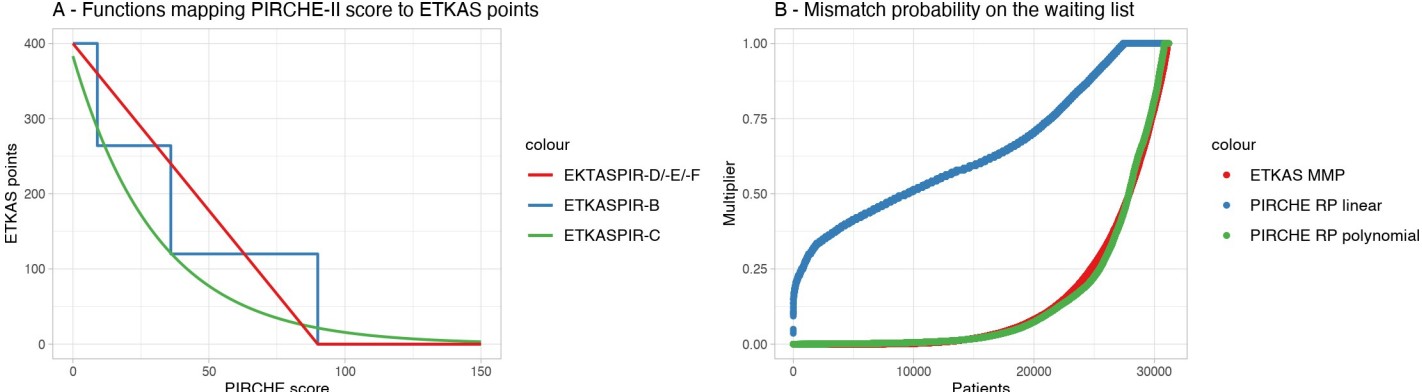

**Fig 3. Match point distributions.** (A) Graphical representation of the distribution of the 400 points for match grade in the PIRCHE-modified models ETKASPIR-B, -C, -D, -E, and -F. (B) Distribution of ETKAS points for HLA MMP on the bootstrapped waiting lists (red curve) shows a benefit for relatively few patients with rare genotypes. Unconverted, the PIRCHE-II risk profile (capped at PIRCHE-II weighted median of 200) assigns more points to average genotypes (blue curve), requiring fitting a polynomial (green curve) to match the regular ETKAS profile.

The distribution of ETKAS points based on the HLA MMP indicates only a limited number of patients benefit from the points. Linear mapping of the PIRCHE-II RP median to ETKAS points revealed however a disproportionate number of patients with a high number of points allocated by the MMP component. Therefore, a fitting polynomial was applied to make sure a similar number of patients benefit from the MMP component with the same proportion (Fig 3B). As suggested by the HLA MMP formula, blood group and PRA were included resulting in following mapping function named PIRCHE-II MMP:

$$\left(1 - bloodgroupFrequency \times relativePRA \times \left(1 - min\left(1, \frac{PIRCHE-II\ Risk\ Profile\ median}{200}\right)\right)\right)^{20}$$

## Evaluation metrics

Validity of the bootstrapping process was evaluated by observing the distributions of patients' blood group, PRA, waiting time, age and country. To evaluate the realism of the baseline simulations, PIRCHE-II distributions were compared to an actual transplant cohort. Additionally, HLA match grade distributions were compared to the baseline simulation.

The benefit of the modified allocation of the status quo simulation was evaluated by calculating the respective areas under the curve (AUC) in previously reported PIRCHE-II ranges knowingly correlated to outcome. PIRCHE-II distributions of simulated transplantations were aggregated by calculating the distribution-weighted mean log(PIRCHE-II) values (WML) in previously defined PIRCHE-II ranges [19]. The SWML aggregates all respective WML scores per simulation. Lower SWML scores indicate more simulated transplantations being carried out with lower PIRCHE-II scores (i.e. lower SMWL is better).

The feasibility of our suggested allocation system changes was evaluated by observing PIRCHE-II scores dependent on HLA match grades and vice versa. Also, the distributions of waiting time of patients on the waiting list and transplanted patients, kidney exchange, country balance, recipient age, waiting list size per country, number of transplantations per country, and points given by histocompatibility were taken into account.

## Statistical analyses

ETKAS-specific data for the years 2013–2017 were collected from the annual Eurotransplant reports as available on the Eurotransplant website [31]. These data were compared with the

bootstrapping data using Student's t-test. The match grade distributions resulting from the simulations were compared by calculating Jensen-Shannon distances and by applying a Wilcoxon signed-rank test to identify significance levels. Evaluations of the allocation performance parameters from the ETKAS simulations and the PIRCHE-II-modified ETKAS simulations were statistically analysed with the Wilcoxon signed-rank test. PIRCHE distributions across the various simulations were compared using the Wilcoxon rank sum test. Estimation of improved graft survival considered reported univariate incidences of graft survival at 10 years post transplant in the respective PIRCHE-II groups [19,20]. PIRCHE-II group frequencies of simulations were multiplied with respective graft survival and summed across all groups. All calculations were executed in R software (R 3.6.1, R Foundation for Statistical Computing, Vienna, Austria).

## Results

### Baseline results after bootstrapping

First, a virtual waiting list was constructed by bootstrapping, using the Eurotransplant Annual Report 2015. The bootstrapping results were categorized according to the strata used in the Eurotransplant Annual Reports and compared to the average of the reported data over the years 2013–1017. Fig 4 shows a comparison of the basic characteristics (A) blood group distribution, (B) panel reactive antibodies (PRA), (C) waiting time since the start of dialysis, (D) age, and (E) country, as reported in the different Eurotransplant Annual reports and after bootstrapping. For the bootstrapping procedure, the waiting time showed a slight increase in number of cases with a waiting time of 0 to 1 year when compared to the ET reports (2159 cases on average in ET versus 2334 cases after bootstrapping; p<0.01). For blood group distribution, PRA, age, and country, no significant differences were observed. These data suggest that the basic characteristics of our initial virtual waiting lists were in general comparable to the basic characteristics of the different Eurotransplant Annual reports, but that the waiting time might be slightly underestimated in our simulations. Yet, the absolute differences were marginal and the individual data points were close to the 95% interval border. Moreover, since the observed differences equally affect all subsequent modified simulations, this deviation was acceptable.

### Validation of the ETKAS simulation model

After performing the ETKAS simulations, we compared the longitudinal ETKAS simulation results with the retrospective outcome parameters as described by Eurotransplant [31]. Fig 5 depicts the longitudinal distributions of the HLA match grades of the Markov chain Monte Carlo (MCMC) simulated ETKAS for a ten-year period. The analyses showed a burn-in period of 2–3 years, after which the match grade distribution stabilized (Fig 5A). From this time point onwards, the simulated ETKAS data resembled the reported data (Fig 5B), although the results in the simulated ETKAS are more stable over the years, whereas the reported data contain more fluctuations.

We next compared the waiting list sizes per country. To this end, the waiting list size data derived from the ETKAS simulations were calculated and plotted over time per country (Fig 5C). For most countries, a slight but constant increase or decrease in waiting list size was observed. These effects extended into all simulated years. The constant increases and decreases are the result of the data used for constructing the model, being the 2015 ET annual report. When comparing the ET report for this year with the 2014 report, the changes in waiting list size compared to the previous year were similar to the observed changes. Such modeling effects were also observed for the country balance, where positive and negative balances from the

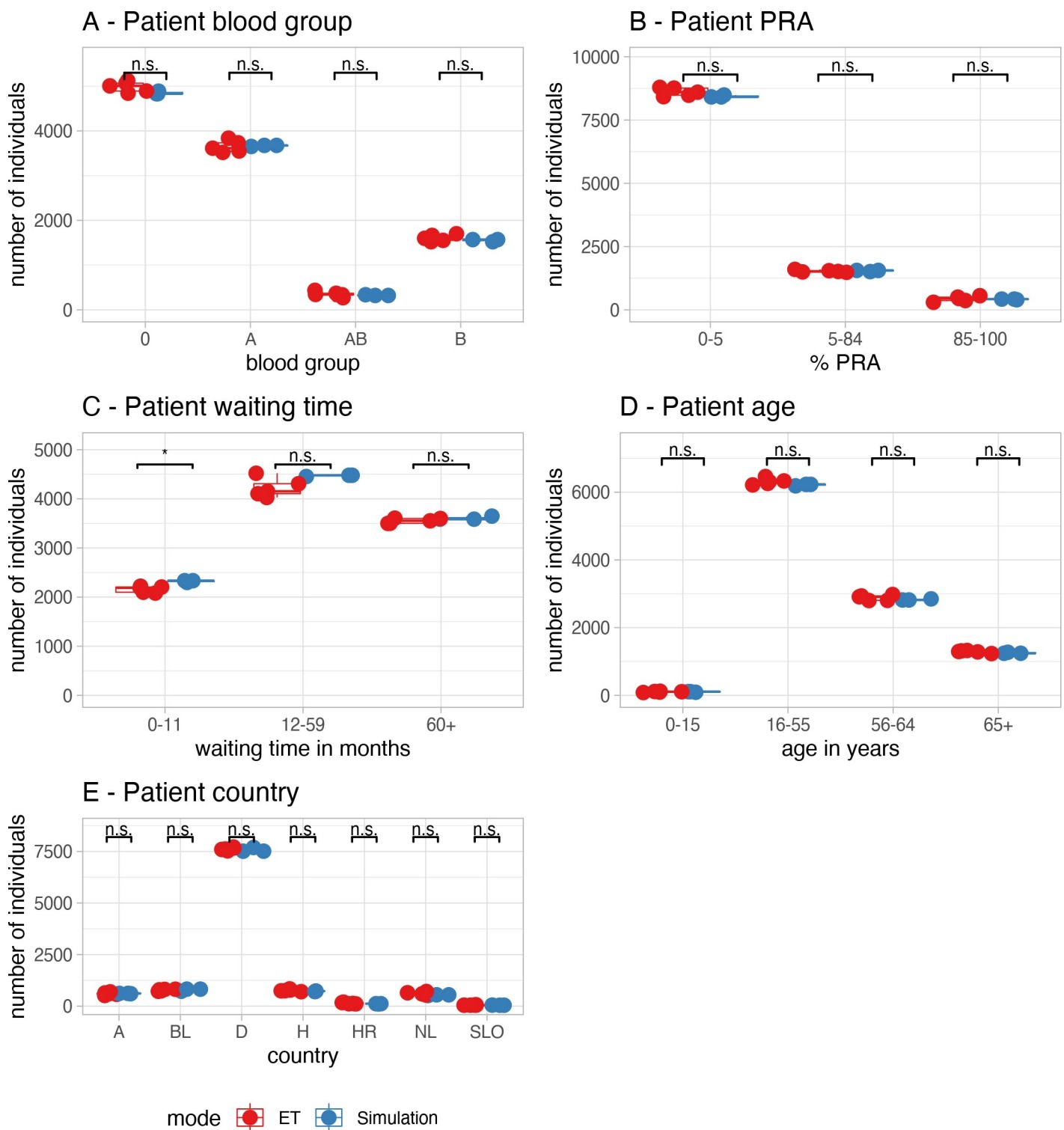

**Fig 4. Bootstrap validation.** Comparison of the bootstrapping results (blue dots) versus the data reported by ET in the years 2013–1017 (red dots). Waiting time was calculated from the start of dialysis. Significant differences (p < 0.01) are indicated with a "*".

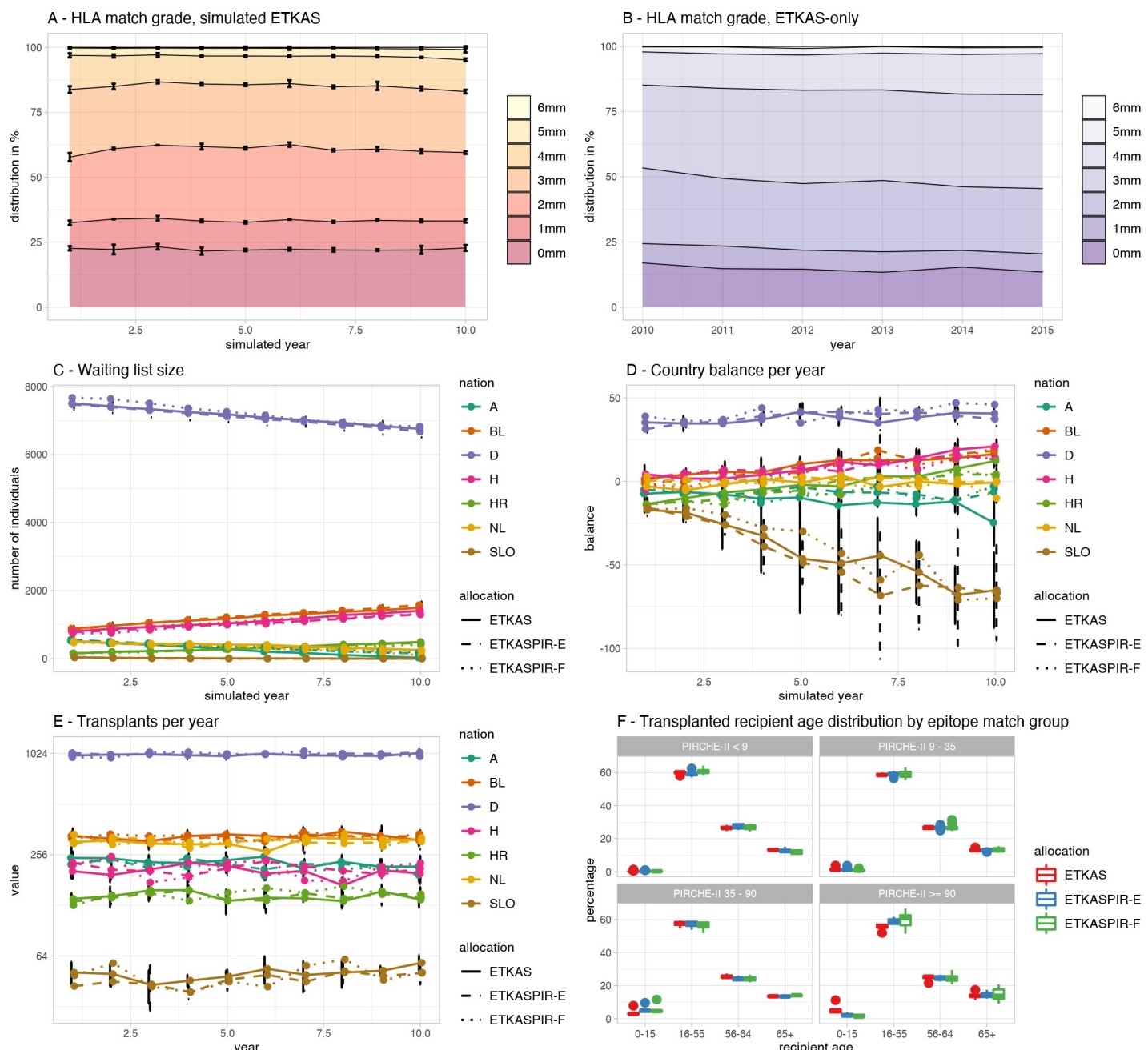

**Fig 5. Simulation stability evaluation.** (A) HLA match grade of the ETKAS simulations, year 10 compared to the 1998 observations: JSD = 0.069, p<0.001, year 10 compared to 2015 ETKAS observations: JSD = 0.135, p<0.001. (B) Observed HLA match grade reported by ET annual reports, only ETKAS. (C) Course of the distribution of the countries of origin of virtual individuals. (D) Course of country balance depending on country and simulated allocation models. (E) Number of transplants per country remained stable between the simulated allocations. (F) distribution of recipient age in the individual PIRCHE groups of different simulation allocations.

2015 report were maintained in the simulations over time (Fig 5D). An exception to these latter observations is Slovenia, where a continuous net export was detected over time. This observation is the result of the low number of patients being registered to the waiting list (61 patients per year) and the comparatively high number of donors offered by this country (43 used kidney donors per year) in 2015. These low numbers result in highly diverse outcomes in

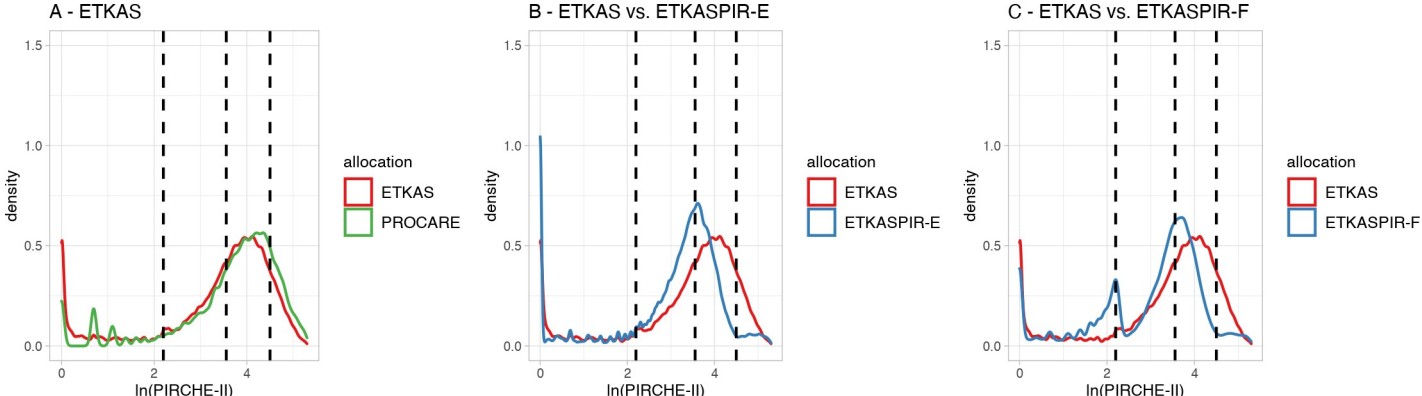

**Fig 6. PIRCHE-II score distributions in simulations.** The log-transformed PIRCHE-II score distributions resulting from the different ETKAS simulation models. Density plots for the simulations were generated using a Gaussian smoothing kernel. Red lines in graphs A-C represent the PIRCHE-II scores in the ETKAS baseline simulations, the green line in graph A represents the previously reported PIRCHE-II scores as observed in the PROCARE cohort[20], blue lines in graphs B-C represent the PIRCHE-II scores in the PIRCHE-II modified ETKAS simulations for the ETKASPIR-E with priority by HLA match, assignment of points linearly descending, and inclusion of a PIRCHE-II based mismatch probability (MMP) (B), and ETKASPIR-F with priority allocation by PIRCHE-II score, assignment of points linearly descending, and inclusion of a PIRCHE-based MMP (C). Dashed lines indicate the previously reported strata by Lachmann et al. [19].

repeated simulations, as indicated by the magnitude of the error bars for Slovenia. In the different allocation simulations, there was no impact on the transplanted recipients' age observed considering the individual PIRCHE-II score groups (Fig 5F).

We next addressed the accurateness of the PIRCHE-II scores in the simulated allocations by comparing them with retrospective real-life allocation data from the previously analyzed kidney transplant PROCARE cohort [20]. The overlay histogram (Fig 6A) shows that the distribution of PIRCHE-II scores in the PROCARE cohort does essentially not deviate from the scores obtained from the ETKAS simulations. As the PROCARE cohort was allocated by ETKAS, our baseline analyses indicate that the primary HLA aspects from ETKAS have been implemented correctly into the ETKAS simulation model.

In conclusion, the validation analyses of the ETKAS simulation model indicate that the simulations are essentially comparable to the Eurotransplant reports, although there are small differences in the details. These subtle differences, however, will be present in all simulation models to the same extent. As such, the ETKAS simulations were used as a baseline for the comparisons with the modified ETKAS simulations below.

## Construction of an improved allocation model (ETKASPIR-E)

The modified allocation approaches ETKASPIR-A to -D were evaluated with their individual components' impact on allocation as demonstrated in S1 Text. Based on these findings, ETKASPIR-E was constructed. Compared to the baseline model, ETKASPIR-E increases the number of transplantations being carried out with lower PIRCHE-II scores (Fig 6B). The number of transplantations with very low PIRCHE-II scores is only slightly increased over ETKAS (17.17% vs. 15.52%; Table 2). However, due to more transplantations being carried out with intermediate-low PIRCHE-II scores (41.66% vs. 25.08%, Table 2) and a therefore decreased number of transplantations with intermediate-high and high scores, the SWML reduces significantly to 3.083 compared to 3.391 of ETKAS (p < 0.001).

## Construction of an optimal allocation model (ETKASPIR-F)

The PIRCHE-II-based prioritization in ETKASPIR-A led to a significant overall improvement as reflected by the proportion of allocations in the lowest risk group and a lower SWML score

**Table 2. PIRCHE-II distribution per risk strata in percent.**

| | | PIRCHE-II score ranges | | | | |
|---|---|---|---|---|---|---|
| | | Group 1 0–9 (WML) | Group 2 9–35 (WML) | Group 3 35–90 (WML) | Group 4 > 90 (WML) | p value (SWML) |
| Allocation model | PROCARE | 10.87% (0.095) | 21.62% (0.672) | 47.71% (1.937) | 19.81% (0.953) | < 0.001[I] (3.657) |
| | ETKAS | 15.52% (0.112) | 25.08% (0.784) | 46.30% (1.871) | 13.10% (0.624) | - (3.391) |
| | ETKASPIR-E | 17.17% (0.116) | 41.66% (1.308) | 36.13% (1.404) | 5.04% (0.256) | <0.001[I] (3.083) |
| | ETKASPIR-F | 27.12% (0.353) | 30.93% (0.993) | 37.02% (1.444) | 4.92% (0.244) | 0.001[II] (3.034) |

Pairwise comparisons of PIRCHE-II scores using Wilcoxon's rank sum test considered (I) ETKAS, (II) ETKASPIR-E. Distribution-weighted mean log(PIRCHE-II) per PIRCHE-II range (WML) in parentheses. Sum of distribution-weighted mean log(PIRCHE-II) values (SWML) integrate frequency and PIRCHE-II distribution.

(S1 Text). The ETKASPIR-D and -E models showed that implementation of a PIRCHE-II-based matching score and mismatch probability can further reduce the overall PIRCHE-II load of the cohort. In the ETKASPIR-F model, we combined all three aspects and evaluated the effects of this combination.

The data in Table 2 show that this combined PIRCHE-II allocation model leads to the optimal projected reduction of risk, as reflected by the SWML score for the ETKASPIR-F model (SWML = 3.034; Table 2). When comparing the proportions and WML scores of the 4 risk groups, ETKASPIR-F benefits from the PIRCHE-II based prioritization. Moreover, this maximisation is not accompanied with an increase of the frequency of allocations into groups 3 and 4. We thus conclude that the ETKASPIR-F model is the most optimal model.

## The effects of PIRCHE-based allocation on other factors

The ETKASPIR-E and -F model yielded the most optimal outcome in the applied MCMC simulations (Table 2). To evaluate the effect of the model on additional allocation outcome parameters, we analyzed these parameters in the ETKASPIR-E model in triplicate and on a single simulation for ETKASPIR-F. All primary comparisons were made between the triplicate ETKAS simulation data and the triplicate ETKASPIR-E data. ETKASPIR-F data were compared to evaluate to confirm identity to the ETKASPIR-E model for these parameters.

**Waiting list characteristics.** In all simulations, all kidneys could be allocated successfully. The simulations showed a burn-in period of approximately 2 years until border-crossing kidney exchange (Fig 7A) was in balance and remained stable for the duration of the simulations. Waiting time distributions of transplanted patients show some fluctuations over the years with mildly decreased waiting time in simulated transplantations with ETKASPIR-E and ETKASPIR-F compared to ETKAS in most years (Fig 7B). Waiting time distributions of patients on the waiting list increased over the years in all models and stabilized after 5 simulated years. ETKAS and ETKASPIR-E showed no statistically significant difference in waiting time, whereas waiting times were slightly increased on the ETKASPIR-F waiting lists in some years (Fig 7C). The number of transplants per country (Fig 4E), the waiting list size per country (Fig 4C) and the country balance per year (Fig 4D) in ETKASPIR-E overlapped with the ETKAS simulations.

**Increased number of HLA mismatches with ETKASPIR-E.** The percentage of 0 mismatch transplants reduced negligibly in ETKASPIR-E compared to ETKAS (means 21.5% versus 22.1%, respectively; Fig 8A). The ETKASPIR-A and ETKASPIR-F models significantly reduce the fraction of 0-mismatched transplants to 12.1% and 11.9% respectively. For the latter two, this is counterbalanced by an increase in the single mismatched transplant group. As the ETKASPIR-B, -C, -D and -E models maintain the ETKAS full match prioritization while the

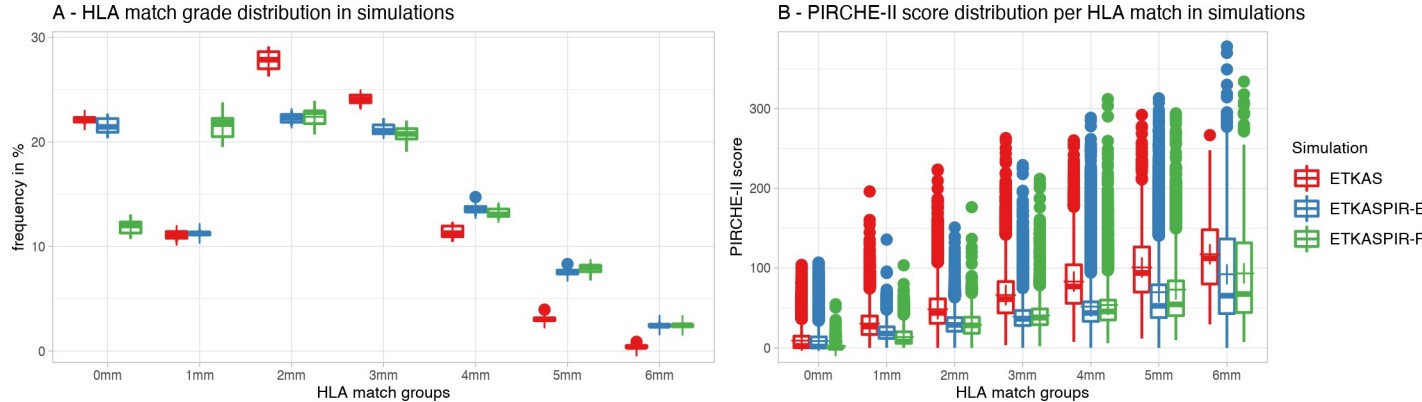

**Fig 7. Monitoring metrics.** (A) After the simulation stabilized, the amount of cross-border donor exchange remained similar between the simulated ETKAS and ETKASPIR-E. Outliers in the burn-in period were suppressed for readability. (B) Average waiting times of transplanted patients was slightly lower in ETKASPIR-E, (C) whereas average waiting times of patients on the waiting list increased mildly. (D) number of match points given by match compatibility in the different simulations. Boxplots depict the median (horizontal line), mean (plus) and first to third quartile (box), the highest and lowest value within 1.5x IQR (whiskers), outliers (circles).

-A and -F models use the PIRCHE-based prioritization, these effects are attributed to the prioritization. These shifts in match grades (Fig 8B), however, still led to a significantly improved PIRCHE-II score for the respective groups as reflected by their SWML scores (Table 2). Thus,

**Fig 8. Impact on match grade.** (A) In ETKASPIR-F (green), the proportion of simulated transplants with a full HLA match reduces when compared to ETKAS (red). ETKASPIR-E (blue) compensates for that by maintaining the current ETKAS priority for HLA full matches. (B) PIRCHE-II scores are reduced in every HLA match group. Boxplots depict the median (horizontal line), mean (plus) and first to third quartile (box), the highest and lowest value within 1.5x IQR (whiskers), outliers (circles).

despite a lower match grade, the PIRCHE-II based matching improves and a better transplant outcome is anticipated for the entire transplant cohort.

For the PIRCHE-based ETKASPIR-B, -C, -D, -E, and -F models, the number of transplants with 2 and 3 mismatches decreased slightly when compared to the baseline ETKAS simulations. Moreover, the number of 4, 5 and 6 mismatched transplants increased when including PIRCHE-II in the allocation match grade algorithm. Again, these increases in mismatches were accompanied by significant decreases in PIRCHE-II scores. Of all factors, replacing the regular match grade scoring system by a PIRCHE-II based scoring resulted in the strongest decrease in SWML score and thus is anticipated to result in the best transplant outcome.

Inclusion of the PIRCHE-II RP in favour of the MMP—to compensate for waiting time effects—had no effect on the mismatch distributions (models ETKASPIR-D versus -E).

**International exchange.** For the ETKASPIR-E and ETKASPIR-F models, we evaluated the effect of these adaptations on the international exchange of kidneys. All models displayed a stabilization period of 4 to 5 years (Fig 7A). After that, the ETKAS and ETKASPIR-E models displayed a similar percentage of organs that were crossing the national borders. These percentages were significantly higher for the ETKASPIR-F model. The differences of the ETKASPIR-F model with the other two were in the range of ~2–3% ($p < 0.001$ for all time points in the range of 5–10 years).

## Discussion

Epitope matching can significantly improve the outcome of HLA-mismatched kidney transplantation [19,20,22,23,32]. Therefore, allocation based on matching for epitope compatibility rather than matching for serological antigen may be beneficial. In contrast to the classical HLA mismatching approaches, which simply count the number of HLA mismatches between a donor and a recipient, epitope matching considers the immunological entities that underlie a potential antibody and T-cell response. These epitopes are derived from the HLA protein sequences and one HLA protein can contain many epitopes. Thus, for epitopes matching, algorithms need to compare the donor's and the recipient's using their amino acid sequences. Consequently, the inclusion of epitope matching factors in the allocation systems would add a novel and complex mathematical layer to the allocation systems. This level of complexity requires computer simulations and cannot be done by intuition. In this study we therefore for the first time addressed the feasibility of including T-cell epitope matching in the organ allocation procedures. Our simulations show that inclusion of PIRCHE-II-based parameters, replacing the classical HLA allocation factors, is feasible without major drawbacks and that it could lead to an improved graft survival equivalent to approximately one extra classical HLA match.

Applying computer simulations to estimate the impact of changes in organ allocation algorithms is of great value. Hence there are many reports on simulation frameworks highly tailored for the respective allocation system they were designed for [33–35]. Also, the introduction of ETKAS by Eurotransplant was preceded by simulating the proposed model [27]. Although very basic to the current standards, these first simulations showed that HLA matching could be included in the allocation in order to achieve an increase of 7%-12% one-year graft survival. The results from these simulations paved the way for including HLA matching in the allocation. Follow up one year after the introduction of the changed allocation procedure showed that these simulations were pretty exact [36]. Similarly, introducing epitope matching techniques to the current allocation system requires thorough monitoring of the expected clinical impact by e.g. monitoring DSA incidence and the respective epitope scores. Considering the reported univariate graft survival rates of previous studies [19,20] and the increasing frequency of patients being transplanted with lower PIRCHE scores, our

simulations of the Eurotransplant Kidney Allocation System suggest an increasing absolute incidence of a functioning graft 10 years after transplantation by 1.1% to 1.5%. This absolute increase translates into a relative improvement of 12.9% to 13.5% bearing in mind the spread in graft survival at 10 years between low and high scores being 8% to 12%. However, confounding factors outside the scope of our simulations may affect these estimates, necessitating further modeling studies.

Beyond tweaking the histocompatibility of the allocation, computer simulations are also capable of evaluating more fundamental characteristics of allocation systems. In 2004 French allocation areas switched from a local center-based allocation policy to a regional patient-centered allocation based on previous simulations showing an improved equity and efficacy [34]. In the United Network for Organ Sharing, computer simulations have a long history with the UNOS Kidney Allocation Model software (UKAM) [33], its successor UNOS Kidney-Pancreas Simulation Allocation Model (KPSAM) [37] and newly suggested models [38,39]. The KPSAM software considers patient and donor data from the Scientific Registry of Transplant Recipients (SRTR) and was instrumental in the 2014 introduction of the kidney donor profile index (KDPI) by the Organ Procurement and Transplantation Network (OPTN) into a new national deceased donor kidney allocation [40]. These simulations predicted that the new policy could potentially improve kidney transplant outcomes with an improvement of 7.0% in median patient life-years per transplant and an average of 2.8% increase in allograft years of life. After the implementation and validation of the current OPTN kidney allocation, KPSAM is still beneficial to evaluate further options to improve utility and equity of kidney allocation e.g. for highly-sensitized recipients [41]. Consequently, SRTR data and simulated allocation models are also applied in lung and liver transplantation, with each setting having its unique allocation system requirements [42,43]. These studies cumulatively show that simulations are a valuable approach to estimate the effects of changes in complex organ allocation algorithms. Although simulations can provide realistic indications regarding the effects on changes to the allocation system, it is crucial to critically evaluate, after implementing changes suggested by computer simulations, if the desired effects were achieved and if potential side-effects were not discovered by previous simulations [36,44]. Such evaluations may be helpful in correcting undesired effects that could not be deduced from the simulations.

Focusing on very specific aspects of the highly complex ETKAS, our allocation models are highly tailored to Eurotransplant's kidney allocation. Nevertheless, basic concepts apply to other transplantation networks in other regions likewise (e.g. virtual populations, patient flow, waiting list management, etc.) and may therefore be reused and adapted accordingly. Also, our simulations may serve as a basis to evaluate more profound changes of the allocation paradigms with respect to utility and equity. Given the variability of organ allocation rules in different countries, adapting simulations or providing a general simulation framework remains however challenging.

Our simulations used the ET kidney transplant data from 2015 to establish a basic ETKAS simulation model. Results from the simulations were comparable with reported data from 2013–2017, indicating the relative exactness of the simulation model. The results show that the simulation is stable over time for HLA matches and number of transplants. These stable results are essential to conclude that the simulated allocation system delivers a long-term reliability and that the simulations do not overrate short-term positive influences. The major differences between our reported historic data from Eurotransplant and our basic ETKAS simulations are the continuous increasing or decreasing trends in the simulation model, for instance for waiting times, waiting list size, and country balance. These trends are caused by the static parameter initialization of the model; conditional probabilities as provided by the reference ET data from 2015 [11] are imbalanced within the observation periods of one year. Consequently, the

specific allocation characteristics in 2015 result in a propagation of bias over the simulation period. In reality, parameters like the number of donors or the number of waiting-list registrations will change over time and result in a balance over longer observation periods. Such aspects were not included in the model to avoid adding uncertain assumptions of distributions to the models. Since these structural imbalances are identical for all simulation models, the modified ETKAS models that include PIRCHE-II-based parameters were compared with the simulated ETKAS model.

Despite the fact that simulations can reflect real-life situations, there may be additional factors that affect the realism of simulation outcomes. For instance, our simulations focused on ETKAS and excluded any interaction with competing or interfering allocation programs within ET, such as the Eurotransplant Senior Program (ESP) for patients over 65 years [45] and the Acceptable Mismatch (AM) Program, for highly sensitized patients [46]. Particularly the AM Program may take specific donors from the regular ETKAS due to prioritization for AM's. This may for instance lead to a lower proportion of homozygous donors for ETKAS, as these donors have by definition a lower number of mismatched antigens and are thus more prone to fit an AM patient than a fully heterozygous donor. Such competing allocation schemes and their interactions are difficult to model and require real-life data of the recipients and the donors, particularly in the case of the AM Program. Similarly, the status of patients on the waiting list is dynamic between "active" (transplantable, T) and "inactive" ("not-transplantable", NT). A substantial proportion of the patients at the waiting list has the status NT and are thus not available for allocation. Data on this dynamic aspect are not available and may affect the simulations. In our simulations we did not consider NT patients, which might underestimate average match grades as the number of distinct patients over time is higher than the number of T listed patients at any given time on the waiting list. It must be acknowledged that changing matching rules dependent on individuals' genotypes may cause advantages/disadvantages for certain subpopulations. These advantages and/or disadvantages may lead to an increase in certain subpopulations on the waiting list. To promote more equality for such situations, Eurotransplant implemented the MMP–a correction factor that adds additional points to patients that are less likely to receive a well-matched transplant. Although we suggest the epitope-equivalent PIRCHE-II RP mechanism over the MMP to promote patients with rare genotypes, further studies are warranted to investigate a potential disadvantage for certain patient groups or populations introduced by epitope matching, e.g. by considering real-world donor and recipient typing data. Other factors that were absent or incomplete in our simulations are the increasing genetic diversity in patients and donors over time for instance due to migration, temporal aspects of immunization, specific center policies, rejected graft offers due to positive crossmatches, organ quality, accuracy of demographic parameters, deregistrations dependent on specific factors, dependency of HLA type and blood group, and the spatial resolution as now defined on country level rather than region. These inaccuracies limit comparability between simulation and real data. However, as we applied a baseline simulation to compare our modified allocation models, the error is systematic and applies similarly to all simulated instances. Such unknown factors could to some extent be included in a simulation model. It is however known that such an approach is prone to result in overfitting the model [47]. We therefore suggest that such data could only be included based upon real-life allocation data as stored by the various allocation organisations, considering the difficulties to apply Markov models to real patient histories. [48–50] With sufficient real-life data of complex, highly-sensitized recipients, simulations could also be extended to estimate the impact of the AM program, which intentionally was excluded in our simulation with virtual individuals. A collaborative project was initiated with Eurotransplant to apply comparable simulations of

ETKAS to their retrospective transplant cohort. This project may be extended to cover also the AM allocation.

Our data indicate that further improvements might be possible. For instance, the optimal ETKASPIR-F model shows that inclusion of the PIRCHE-II based score may reduce the assigned HLA-related scores (Fig 7D); distribution of 400 points may be improved further possibly by optimizing the distribution curve. Such improvement might be established by adding a 400-point plateau for the lower group or distributing these points on a linear basis or via an arccotangent-based function. Second, for some simulated ETKAS-PIR models, the waiting time for transplanted patients is reduced, while patients wait longer on the waiting list. This phenomenon can be explained by the average number of points given by the HLA and respectively PIRCHE match component. As the number of points reduces on average, the burden of overcoming a low histocompatibility with points derived from waiting time is lower, requiring less waiting time. Correction of these effects may be achieved by applying an adjustment factor to the HLA component, so the average number of points given matches the current ETKAS implementation.

Our simulations considered PIRCHE-II scores with HLA-A, -B, -C, -DRB1 and -DQB1 as peptide sources, with HLA-DRB1 as the only presenting locus. With multiple imputation enabled, providing HLA-A, -B and -DRB1 is sufficient to calculate reasonably accurate PIRCHE-II scores. [51] This matches the requirements of the current Eurotransplant allocation rules, potentially reducing the burden of implementing suggested changes. However, there is growing evidence of the clinical relevance of presentation of allopeptides by e.g. HLA-DQA1/-DQB1 heterodimers to CD4+ T cells. [22] After further clinical validation, implementation of corresponding typing methods in the pre-transplant diagnostic routine and upon availability of haplotype frequency datasets including HLA-DQA1 and HLA-DP, the presented workflow might be adapted to simulate enhanced PIRCHE-II scores' impact on allocation.

The implementation of PIRCHE-II based allocation is projected to improve transplant outcome. This implementation additionally increased the international exchange of organs. International exchange within Eurotransplant has not been documented in the annual reports. As such, the validity of our simulations could not be evaluated. This exchange aspect in the context of HLA matching was, however, recognized at the time of implementation of HLA matching for kidney transplantation and is one of the key benefits of international organ allocation organizations as Eurotransplant [36]. In our simulation analyses, international exchange rates increased in those models where low PIRCHE-II scores were prioritized (ETKAS versus ETKASPIR-F), but not when prioritization was still based upon full HLA match (model ETKASPIR-E versus -F). An increasing exchange might indicate longer average cold-ischemic times, which impairs outcome. If desired, such effects could in principle be counterbalanced by changing the cutoff for prioritization or by adding more weight to the non-HLA factor for national allocation. The latter is, however, outside the scope of the current investigations.

In summary, we here for the first time executed structural and extensive MCMC simulations to evaluate the feasibility of implementing T-cell epitope matching based upon the PIRCHE-II concept in kidney allocation. Our data show that, by allowing more international exchange, such an implementation is viable. The evaluations indicate that the relative benefit in graft survival is around 13%. A better epitope matching is also expected to reduce the immunisation rate including a lower incidence of dnDSA formation [19]. The latter will reduce the number of unacceptable antigens in a potential subsequent retransplantation. Collectively, the cumulative graft years are likely to increase, patients will return to the waiting list later than in the current system and overall waiting list size will shorten or grow at a slower rate in the long run.

## Supporting information

**S1 Text. Evaluation of ETKASPIR-A—ETKASPIR-D.**
(DOCX)

## Author Contributions

**Conceptualization:** Matthias Niemann, Eric Spierings.

**Data curation:** Matthias Niemann.

**Formal analysis:** Matthias Niemann.

**Investigation:** Matthias Niemann, Eric Spierings.

**Methodology:** Matthias Niemann, Eric Spierings.

**Software:** Matthias Niemann.

**Supervision:** Eric Spierings.

**Validation:** Matthias Niemann, Eric Spierings.

**Visualization:** Matthias Niemann.

**Writing – original draft:** Matthias Niemann, Eric Spierings.

**Writing – review & editing:** Matthias Niemann, Nils Lachmann, Kirsten Geneugelijk, Eric Spierings.

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
