## [Decision Letter · Decision Letter 0]

1 Apr 2021

Dear Mr. Niemann,

Thank you very much for submitting your manuscript "Computational Eurotransplant Kidney Allocation simulations demonstrate the feasibility and benefit of T-cell epitope matching" for consideration at PLOS Computational Biology.

As with all papers reviewed by the journal, your manuscript was reviewed by members of the editorial board and by several independent reviewers. In light of the reviews (below this email), we would like to invite the resubmission of a significantly-revised version that takes into account the reviewers' comments.

We cannot make any decision about publication until we have seen the revised manuscript and your response to the reviewers' comments. Your revised manuscript is also likely to be sent to reviewers for further evaluation.

Sincerely,

Roeland M.H. Merks, Ph.D

Associate Editor

PLOS Computational Biology

Mark Alber

Deputy Editor

PLOS Computational Biology

Reviewer's Responses to Questions

**Comments to the Authors:**

Reviewer #1: "First, a virtual waiting list was constructed by bootstrapping, using the Eurotransplant Annual Report 2015." You need to provide a great deal more information on how you performed your simulation, since all of your conclusions hinge on it. Since the Eurotransplant Annual Report provided no individual patient data, how did you 'bootstrap' a waitlist? Bootstrapping usually involves constructing random samples with replacement of a set of individual observations. How did you manufacture counterfactual patient histories? For example, suppose a patient had originally received a transplant in March, but your simulation did not give them a transplant until December (if at all). How do you know the patient is still alive? If you did not construct these histories, then it is a foregone conclusion that your approach would appear (in your simulations) to have better outcomes; patients who never received a transplant under your allocation system would be immortal. Note: please do not use Markov models to simulate patient histories, as patient medical histories completely violate the 'memoryless' assumption that Markov models depend on.

"Subsequently, for the non-HLA typing characteristics, the parameters country, urgency, recipient age, blood group,

percentage of PRA, and waiting time were extracted from the Eurotransplant Annual Report 2015 39 and assigned to the virtual waiting list population accordingly. Country, urgency and age were considered as independent parameters and thus assigned randomly, using the reported overall frequencies. Distributions for blood group, PRA, and waiting time were considered conditional to the virtual individuals’ country and thus assigned according to country-dependent frequencies. " This contradicts the above statement, and implies that you assigned random ages, blood groups, PRA percentages, and waiting times. How did you handle the inter-dependent factors? Women are more likely to be older, and they're more likely to be highly sensitized. Older patients are less likely to survive long periods of time on the waitlist. How does blood group relate to HLA? Generally speaking, it is not a good idea to consider two demographic or medical variables to be completely independent of each other.

"our simulations of the Eurotransplant Kidney Allocation System suggest an increasing absolute incidence of a functioning graft 10 years after transplantation by 1.1% to 1.5%. This absolute increase translates into a relative improvement of 12.9% to 13.5% bearing in mind the spread in graft survival at 10 years between low and high scores being 8% to 12%"

You do not describe how you estimated this survival improvement, although you imply it's based solely on matching. Kidney and patient survival is affected by a great deal more than just antigen and epitope matching. Donor and recipient demographics (especially age) comorbidities and other case mix factors all have substantial impact on patient and graft survival. Any statement that a new allocation system would improve survival needs to do more than just look at matching. Does epitope matching affect racial allocation? If so, then changing the allocation may be affecting disparities in racial access to transplantation, and thus survival. Similarly, if patients had to wait longer for kidneys, then the increased dialysis time is likely to reduce post-transplant survival.

"We first performed an allocation in triplicate using the unmodified ETKAS procedure as described in the Eurotransplant ETKAS Manual fig 7." Your statement that you used an unmodified ETKAS allocation system is incorrect. You omitted the ETKAS allocation rules for highly-sensitized candidates. Why? This is an important group that would likely be strongly impacted by changes in matching allocation rules.

Would your allocation system affect the blood-type distribution of recipients? O-type recipients are already disadvantaged; would your system exacerbate this?

Did your allocation system use all the kidneys, or were there changes in whether some of the marginal kidneys were used or discarded?

Your simulation references are fairly incomplete. There has been a fair amount of work done in the past twenty years to develop adequate organ allocation simulations. Please review more of the literature. Some of the American work done by the SRTR might be helpful.

Reviewer #2: Thank you for your very interesting and promising paper. In its current presentation, the result section is quite difficult to read and to understand. Reading the results section before the methods section make the results and their relevance very difficult to understand indeed. This make you paper falsely looking more a technical report than a scientific one.

If one consider the title, your are supposed to demonstrate the feasibility and the benefit of integrating the T-Cell epitope matching within ETKAS.

1) For this purpose, do you think it is really necessary to have results for 8 allocations models including 6 simulated ones ? What does PROCARE add to the demonstration ? Of course ETKAS is required but I suggest you to select the minimum of simulations required to the demonstration. You could explain you had a stepwise approach whose results are provided as supplementary material.

2) Gaining place, you could then explain more the evaluation metrics you are using, especially in terms of the feasibility and benefit you are supposed to demonstrate. What are metrics to challenge the realism of the simulation (data generation, process, allocation scheme). What are the metrics used to validate the feasibility and those for the benefit.

3) I suggest also you to include ETKAS, Substituting HLA matching by PIRCHE-II based matching, Full HLA match prioritization, HLA match grade replacements and HLA mismatch probability sub-sections before the presentation of the results which are quite incomprehensible without.

4) Anyway, it would be better to have the Material and Methods before the results but I guess this is possible for PLOS computational biology. This section should also separate more clearly what is related to data generation, process simulation (MCMC) and allocation scheme simulation which are the three pillars of you entire "simulation" work.

5) The introduction is clear, maybe should you end with what can be expected in terms of feasibility and benefit for ETKAS. It could also starts with the fact that we are in the context of Kidney transplantation.

6) Of note, some of bibliographical references in the text goes to strange references according to the statement in the text and the paper it refers to. Could you check this.

7) The discussion is also very interesting. I suggest you state as an underlying but very reasonable hypothesis that epitope matching is better than actual serological matching. I found pretty smart the way you generated Ab in the recipients (forbidden antigens). But for many immunologists it looks that the number of eplets related to an epitope might be misleading. What matters is the 3D conformation of HLA molecule. To avoid the apparition of de novo DSA after KTx is according to epitope mismatch is the real objective, that needs a prospective evaluation. In the introduction, you state that "recent studies have shown that predicted donor-HLA epitopes presented on HAL class-II molecules ... are related to HLA antibody formation. I suggest you add the references to theses studies and move this point in the discussion. Any future works ? What about reproducibility of in silico experimentations in other countries ?

**Have all data underlying the figures and results presented in the manuscript been provided?**

Reviewer #1: Yes

Reviewer #2: Yes

PLOS authors have the option to publish the peer review history of their article (what does this mean?). If published, this will include your full peer review and any attached files.

Reviewer #1: No

Reviewer #2: No
---

## [Decision Letter · Decision Letter 1]

19 Jun 2021

Dear Mr. Niemann,

Thank you very much for submitting your manuscript "Computational Eurotransplant Kidney Allocation simulations demonstrate the feasibility and benefit of T-cell epitope matching" for consideration at PLOS Computational Biology. As with all papers reviewed by the journal, your manuscript was reviewed by members of the editorial board and by several independent reviewers. The reviewers appreciated the attention to an important topic. Based on the reviews, we are likely to accept this manuscript for publication, providing that you modify the manuscript according to the review recommendations.

Sincerely,

Roeland M.H. Merks, Ph.D

Associate Editor

PLOS Computational Biology

Mark Alber

Deputy Editor

PLOS Computational Biology

[LINK]

Reviewer's Responses to Questions

**Comments to the Authors:**

Reviewer #1: Thank you for further explaining your simulation methods and addressing my previous comments.

You said that "We think it is not to be expected that epitope matching is more prone to “racial allocation”

than plain serologic matching."

Let me explain what I meant. The population on the waitlist is not necessarily racially distributed similarly to those of a marrow donor program. We know that the risk of kidney disease is not equal across races, and that race is linked with haplotype, blood group, and patient survival among kidney disease patients. While all of these may impact the validity of your simulation, what I was referring to was the idea that changes in the allocation system that are related to haplotype distribution may affect the racial distribution of the recipients. I realize you don't have the ability to describe this with your data, but you should mention it as a possibility that should be investigated, unless there's something I'm not understanding.

Reviewer #2: Thank you for the revised version which is really improved. Some minor changes could be added. (1) maybe a reference for Gibbs sampler. (2) I see that DQA is not taken into account. This might be a future work as epitopes for Ab anti DQ are often located on a common zone of alpha and beta chain. DQB matching only might no be ideal or its relevance could be challenged, even at epitaphic level. (3) PIRCHE-II RP : RP acronym = ? (4) In the evaluation metrics section, you could just state for the reader that low SWML score means better matching or not. In the discussion: iI suggest you to replace "there may be additional factors that affect the simulation outcomes" with "there may be additional factors that affect the realism of simulation results". (5) "Transplantable" and "Not Transplantable" could be replaced by "Active" and "Inactive" status.

**Have the authors made all data and (if applicable) computational code underlying the findings in their manuscript fully available?**

Reviewer #1: Yes

Reviewer #2: **No: **Not easily applicable but sharing simulations coding files to make in silico experimentations reproductible by other research teams could be very interesting.

PLOS authors have the option to publish the peer review history of their article (what does this mean?). If published, this will include your full peer review and any attached files.

Reviewer #1: No

Reviewer #2: **Yes: **Christian Jacquelinet

Figure Files:

Data Requirements:

Reproducibility:

References:

---

## [Editor Report · Decision Letter 2]

5 Jul 2021

Dear Mr. Niemann,

We are pleased to inform you that your manuscript 'Computational Eurotransplant Kidney Allocation simulations demonstrate the feasibility and benefit of T-cell epitope matching' has been provisionally accepted for publication in PLOS Computational Biology.

Best regards,

Roeland M.H. Merks, Ph.D

Associate Editor

PLOS Computational Biology

Mark Alber

Deputy Editor

PLOS Computational Biology

---

## [Editor Report · Acceptance letter]

20 Jul 2021

PCOMPBIOL-D-20-02268R2 

Computational Eurotransplant kidney allocation simulations demonstrate the feasibility and benefit of T-cell epitope matching

Dear Dr Niemann,

I am pleased to inform you that your manuscript has been formally accepted for publication in PLOS Computational Biology. Your manuscript is now with our production department and you will be notified of the publication date in due course.

With kind regards,

Katalin Szabo
